# Hysterectomy for Benign Indications and Risk of Cataract Formation in South Korean Women

**DOI:** 10.3390/medicina59091627

**Published:** 2023-09-08

**Authors:** Jae Suk Kim

**Affiliations:** Department of Ophthalmology, Sanggye Paik Hospital, Inje University College of Medicine, Seoul 01757, Republic of Korea; s5535@paik.ac.kr; Tel.: +82-2-950-1101

**Keywords:** adnexal surgery, cataract, hysterectomy, estrogen, risk factor

## Abstract

*Background and Objectives:* Few studies have investigated the relationship between female reproductive hormones, especially estrogen, and the incidence of cataracts. This study sought to evaluate the effects of hysterectomy on the risk of lens opacity in Korean women. *Materials and Methods:* This retrospective cohort study utilized data collected from 2007 to 2020 for 255,576 postmenopausal women in the Korean National Health Insurance database. Participants were divided into those who did and did not undergo hysterectomy. The hysterectomy group was further divided into two subgroups according to the type of adnexal surgery performed. The prevalence of cataracts was then compared among the control, hysterectomy alone, and hysterectomy with adnexal surgery groups. *Results:* The control group included 137,999 participants who did not undergo hysterectomy, while the treatment group included 93,719 women who underwent hysterectomy alone or in combination with adnexal surgery. The incidence of cataracts was higher in the control group than in the treatment group, as demonstrated in a 1:1 propensity score-matching analysis adjusted for potential confounding variables. *Conclusions:* The incidence of cataracts was significantly lower in the group with hysterectomy than in the control group, but the difference was subtle. The current findings may aid in identifying the role of female reproductive hormones in cataract development.

## 1. Introduction

Cataracts are among the major causes of blindness in older adults worldwide [1]. Given the continued increase in the aging population, the number of people diagnosed with cataracts has also increased. Previous studies have reported risk factors for cataract development and progression, such as age, diabetes mellitus, ultraviolet light, hormones, and certain drugs [2,3,4]. Female sex has also been identified as a risk factor for cataracts in previous epidemiologic studies, wherein postmenopausal women exhibited a higher incidence of cataracts than men of similar age [3,4]. Such research suggests that female reproductive hormones play a role in the development of cataracts [4].

Estrogen is one of the most important reproductive hormones in women. Endogenous exposure to female reproductive hormones starts with the release of estrogen from the ovary [5], which is influenced by various factors, including the age of menarche and menopause and history of hysterectomy or adnexal surgery [5,6]. Estrogen receptor mRNAs are present in the retina, ciliary body, iris, and lens epithelial cells [7]. It can be considered that estrogen has a direct effect on the lens. However, results from epidemiologic studies on the relationship between estrogen exposure and the risk of cataracts are limited and have been inconsistent. 

In an experimental rat model, estrogen exerted a protective effect against the apoptosis of lens epithelial cells [8]. Estrogen modulates the immune response by regulating the expression and secretion of inflammatory cytokines [9]. In the formation of cataracts, inflammatory response is one of the key roles, and oxidative stress is one of them [10]. Estrogens have been found to have antioxidative properties [11,12]. Moreover, laboratory studies have indicated that estrogen may protect against the development of cataracts influencing a TGFb-mediated mechanism of cataract formation [13,14]. 

Some studies suggest different effects of estrogen on cataracts. There was a study that showed that progesterone and 17-estradiol induced an increase in Ca2^+^ [15], and a study showed that the increased Ca2^+^ in the lens was associated with cataract generation [16,17,18]. Inflammatory parameters such as C-reactive protein (CRP) may be involved in estrogen and cataract development. Estrogen raises CRP levels as a result of liver synthesis of protein [19,20], and hepatically-induced CRP is associated with cataract development [21]. 

Hysterectomy, one of the most common gynecological surgeries [22], has been associated with a decrease in ovarian reserve [23,24]. While some studies have reported that estrogen exposure is associated with a decrease in the prevalence of cataracts [25,26,27], others have demonstrated no protective effect of endogenous estrogen exposure on cataract development or progression [28,29]. Research regarding the relationship between hysterectomy and cataract formation is lacking, and few studies have adjusted for comorbidities like diabetes mellitus and hypertension, which are known risk factors for cataracts.

To address these issues, the present retrospective cohort study analyzed Korean National Health Insurance data collected from 2007 to 2020 to identify the relationship between hysterectomy and the incidence of cataracts in Korean women.

## 2. Materials and Methods

### 2.1. Database, Study Design, and Ethics

All citizens of South Korea are legally obliged to enroll in the country’s health insurance program, The National Health Insurance Service (NHIS). The NHIS has medical information for 51 million Koreans, such as age, sex, diagnosis, drug prescription, type of surgery, type of medical insurance, hospitalizations, and outpatient examinations. The NHIS shares most of this information with the Health Insurance Review and Assessment Service (HIRA), which was established to examine the suitability of billed medical expenses to prevent disputes between the NHIS and hospitals over insurance payments. This population-based retrospective cohort study analyzed HIRA data collected between January 2007 and December 2020. 

### 2.2. Participant Selection

This study used the International Classification of Diseases, 10th revision, and Korea Health Insurance Medical Care Expenses (2016, 2019 versions) for participant selection and statistical analysis. In this study, the hysterectomy group comprised women aged 40–59 years who underwent hysterectomy for uterine myoma or elective hysterectomy between 1 January 2011 and 31 December 2014. When adnexal surgery (extirpation of adnexal tumor (unilateral or bilateral oophorectomy, unilateral or bilateral salpingoophorectomy, unilateral or bilateral salpingectomy, unilateral or bilateral ovarian cystectomy); incision and drainage of ovarian cyst; ovarian wedge resection; and adhesional adnexectomy) was performed on the same day, the case was defined as hysterectomy with simultaneous adnexal surgery. The hysterectomy date was set as the day of inclusion. 

Cases of any cancer or benign lens disease (H25 (disorders of lens), H26 (other cataract), H27 (other disorders of lens), H28 (cataract and other disorders of lens in diseases classified elsewhere)) diagnosed within 180 days of inclusion were excluded from both groups. Among the hysterectomy and control groups, 1:1 propensity score matching was performed for factors such as age, year of inclusion, socioeconomic status (SES), region, Charlson Comorbidity Index (CCI), parity, adnexal surgery before inclusion, hypertension, diabetes mellitus, dyslipidemia, menopause, and menopausal hormone therapy (MHT) before inclusion.

### 2.3. Outcome 

Patients who visited any hospital with cataract-related diagnostic codes (H25, H26.8, H26.9, and H28.2) more than thrice were considered to have cataracts.

### 2.4. Variables

Participants whose medical insurance was medical assistance were defined as having a low SES. Non-metropolitan regions were defined as rural areas. CCI was obtained in accordance with the method reported by Quan et al. [30], using diagnostic codes reported from the first examination date through 1 year before inclusion. Parity was defined as the number of labors within the study period. Participants who had undergone adnexal surgery more than once before hysterectomy were considered to have a history of adnexal surgery. Hypertension (I10–15); diabetes (E10–14); dyslipidemia (E78); and menopause (N95, N80.0, M81.0, and E28.3) were defined when the participant visited medical institutions more than twice with the corresponding diagnostic code. MHT before inclusion was defined as the first menopausal hormone treatment (tibolone, estradiol valerate, estradiol hemihydrate, dydrogesterone, norethisterone acetate, medroxyprogesterone acetate, drospirenone, and cyproterone) prescribed for more than 180 days before inclusion. MHT after inclusion was defined as menopausal hormone treatment prescribed for the first time after inclusion (>180 days of total prescription).

### 2.5. Statistical Analysis

All statistical analyses were performed using SAS Enterprise Guide 7.15 (SAS Institute Inc., Cary, NC, USA) and supplemented with R 3.5.1 (The R Foundation for Statistical Computing, Vienna, Austria). A *p*-value < 0.05 was considered statistically significant. 

The Cochran–Mantel–Haenszel test was used to analyze categorical variables, and the Wilcoxon signed-rank test was used to analyze continuous variables. Standardized differences were used to evaluate variables subjected to matching. Stratified Cox regression analysis was performed to determine the contribution of hysterectomy to cataract risk after adjusting for confounding factors. The first day for Cox analysis was set as the inclusion day for each group, and the last day was set as the date of the end of the study (31 December 2020). The listwise deletion method was used when the missing value was less than 10%. When the missing value was greater than 10%, a regression imputation method was applied. 

To confirm the robustness of our study, stratified Cox regression analysis was performed on participants who underwent laparotomic hysterectomy to analyze the risk of cataract formation. Continuous variables are expressed as median values (25th percentile, 75th percentile), and categorical variables are expressed as numbers (percentages). Cox proportional hazards models were implemented to adjust for various confounding factors and as a sensitivity test after selecting cases prescribed by an obstetrician in the MHT group. All analyses were performed using two-sided tests. 

### 2.6. Ethics

This study was approved by the Institutional Review Board (IRB) of Sanggye Paik Hospital (approval number: SGPAIK 2021-12-005). Before releasing the medical information to researchers, HIRA removes identifying variables for each patient. Furthermore, HIRA permits analysis only on closed servers, and only the anonymized resultant tables, figures, and numbers can be taken outside the server. Therefore, researchers are fundamentally unable to specify each patient, and there are no possible disadvantages for the participants. In accordance with the Bioethics and Safety Act of South Korea, these participants were not obliged to provide informed consent. Therefore, raw data could not be provided.

## 3. Results

### 3.1. Patient Characteristics

Data were extracted for 255,576 patients who underwent national examinations by the Korean Health Insurance Corporation. Among these patients, 145,856 were classified into the non-hysterectomy group, while 93,179 were classified into the hysterectomy group. 

After 1:1 propensity score matching, 84,757 participants were included in the two groups (i.e., the premenopausal and menopausal groups) (Figure 1). The median age in the non-hysterectomy group was 46 (43–50) years, while the median age in the hysterectomy group was 47 (44–50). In the hysterectomy group, 65,702 (77.5%) patients underwent hysterectomy alone, and 19,055 (22.5%) underwent hysterectomy with adnexal surgery. The additional baseline characteristics of the participants are shown in Table 1.

### 3.2. Hysterectomy and Cataract

Cataract formation was observed in 1787 (2.1%) patients in the non-hysterectomy group and 1648 (1.9%) patients in the hysterectomy group. In the subgroup analysis, the number of patients with cataract formation was 1235 (1.9%) in the hysterectomy alone group and 413 (2.2%) in the hysterectomy with adnexal surgery group (Table 2 and Appendix A). Before adjusting for confounding variables, the hazard ratios for cataract development were 0.904 (0.843–0.968, *p* = 0.004) in the hysterectomy group, 0.924 (0.854–1.002, *p* = 0.058) in the hysterectomy alone group, and 0.844 (0.737–0.967, *p* = 0.015) in the hysterectomy with adnexal surgery group. After adjusting for hysterectomy, age, SES, region, CCI, parity, adnexal surgery, hypertension, diabetes mellitus, dyslipidemia, menopause, and MHT before inclusion, the hazard ratios for cataracts were 0.886 (0.825–0.952, *p* < 0.001) in the hysterectomy group, 0.91 (0.837–0.989, *p* = 0.026) in the hysterectomy alone group, and 0.822 (0.716–0.944, *p* = 0.006) in the hysterectomy with adnexal surgery group. After additional adjustment for MHT after inclusion with previous variables, the hazard ratios were 0.902 (0.838–0.971, *p* = 0.006) in the hysterectomy group, 0.924 (0.849–1.006, *p* = 0.067) in the hysterectomy alone group, and 0.842 (0.732–0.968, *p* = 0.016) in the hysterectomy with adnexal surgery group. 

The hazard ratio of MHT after inclusion for cataract development was 0.852 (0.735–0.988, *p* = 0.034) (Table 3, Figure 2). 

After adjusting for confounding variables, hazard ratios for cataracts according to age group were 0.981 (0.861–1.117, *p* = 0.77) in the hysterectomy group, 0.998 (0.662–1.156, *p* = 0.982) in the hysterectomy alone group, and 0.919 (0.698–1.212, *p* = 0.551) in the hysterectomy with adnexal surgery group among participants aged 45–49 years; 0.845 (0.751–0.95, *p* = 0.005) in the hysterectomy group, 0.883 (0.727–0.955, *p* = 0.009) in the hysterectomy alone group, and 0.877 (0.703–1.093, *p* = 0.242) in the hysterectomy with adnexal surgery group among participants aged 50–54 years; and 0.813 (0.669–0.988, *p* = 0.038) in the hysterectomy group, 0.872 (0.684–1.112, *p* = 0.271) in the hysterectomy alone group, and 0.724 (0.532–0.987, *p* = 0.041) in the hysterectomy with adnexal surgery group among participants aged 55–59 years (Table 4, Appendix A). 

## 4. Discussion

In this large-scale, long-term study of Korean Health Insurance Corporation data for Korean women, we observed that hysterectomy decreased the risk of cataract formation compared to that observed in the control group. Furthermore, patients receiving MHT after hysterectomy were included, although the study participants were relatively young, and these factors were statistically adjusted. After adjusting for several variables (i.e., hysterectomy, age, SES, region, CCI, parity, adnexal surgery before inclusion, hypertension before inclusion, DM before inclusion, dyslipidemia before inclusion, menopause before inclusion, and MHT before inclusion), both hysterectomy alone and hysterectomy with adnexal surgery were shown associated with a lower prevalence of cataracts. 

Estrogens released from the ovary play important roles in the reproductive system [31]. Hysterectomy is one of the most frequently performed gynecological surgeries [22] wherein the blood supply to the ovary may be compromised, which may lead to ovarian failure [32,33]. Even when the ovaries are preserved, the risk of ovarian failure is 15% in women who have undergone a hysterectomy, which is twice the risk in women who do not undergo a hysterectomy [34]. 

Previous studies have reported compromised ovarian function after hysterectomy based on changes in estradiol levels, follicle-stimulating hormone, and anti-Mullerian hormone [24,35]. Estrogen and progesterone receptor mRNAs are present in lens epithelial cells [7]; however, whether estrogen receptors are related to cataract formation remains unknown. In our study, the prevalence of cataracts was lower in the hysterectomy alone group than in the non-hysterectomy group after adjusting for variables except MHT after inclusion. 

Since surgical menopause occurs earlier when adnexal surgery is performed with hysterectomy, the risk of cataract incidence was compared by dividing the cohort into a hysterectomy alone group and a hysterectomy with adnexal surgery group. Adnexal surgery, including any kind of operation (unilateral or bilateral oophorectomy; unilateral or bilateral salpingoophorectomy; unilateral or bilateral salpingectomy; unilateral or bilateral ovarian cystectomy incision and drainage of ovarian cyst; ovarian wedge resection; and adhesional adnexectomy), affects ovarian function. 

Previous studies have indicated that unilateral oophorectomy may cause failure of the contralateral ovary independent of hysterectomy [36,37]. Concurrent hysterectomy may cause premature estrogen deficiency in patients who have undergone unilateral oophorectomy [33]. Farquhar et al. described similar results in patients with menopausal symptoms, reporting that 20% of patients who underwent hysterectomy experienced hot flashes, while 43% of patients who underwent hysterectomy with bilateral oophorectomy experienced hot flashes [38]. In the present study, the hazard ratio was lower for the hysterectomy with adnexal surgery group than for the hysterectomy alone group. It can be considered that surgical menopause has a protective effect on cataract formation. Previously, the majority of studies regarding the cataract formation of early menopause and MHT reported that estrogen had a protective effect on cataract formation [39,40,41,42,43], Considering the lower estrogen levels after hysterectomy with or without adnexal surgery, this study shows a conflicting result with the previous studies. To our knowledge, this is the first study of the effect of surgical menopause on cataract formation.

However, after adjusting for variables including MHT, hysterectomy alone was not associated with the risk of cataracts. Previous studies on the relationship between MHT and cataract risk have reported limited and inconsistent results. In the Beaver Dam Eye Study, MHT was associated with a lower incidence of cataracts, and a later onset of menopause was associated with a protective effect against nuclear sclerosis of the lens [3]. In contrast, in the POLA study, the authors observed no association between MHT use and cataract formation, and there was a greater rate of cataract surgery in current MHT users [28]. In the Blue Mountains Eye Study, MHT was not associated with nuclear sclerosis or cortical/posterior subcapsular cataracts [35]. 

In the present study, MHT after inclusion decreased the risk of cataracts. This result supports the hypothesis of previous studies that estrogen has a protective effect on cataract formation. However, the drug’s type, dose, and duration could not be identified because MHT after inclusion was used as a variable in this study. Additionally, MHT use was significantly more frequent in the hysterectomy group than in the non-hysterectomy group due to the development of surgical menopause after hysterectomy. This study could not verify the effect of MHT as a sole variable on cataract incidence because 1:1 propensity score matching was performed to compare the hysterectomy and non-hysterectomy groups. 

To our knowledge, this is the first large-scale, single-race study on the relationship between hysterectomy and cataract formation. Our study is also advantageous compared to previous studies, given that the data were acquired from the National Health Screening Program and NHIS; thus, this study included participants from all regions of Korea. Furthermore, analyses were adjusted for numerous variables (age, BMI, SES, CCI, parity, age at menarche, age at menopause, smoking, alcohol consumption, physical exercise, and period since menopause), and 1:1 propensity score matching was performed.

This study had several limitations. First, cataract types could not be identified because the cataract diagnostic codes used in this study did not differentiate between cataract types. Second, the effects of the type of adnexal surgery on cataract formation could not be identified because the number of surgeries performed was not available due to the policies of the NHIS. In addition, as “cataract” was identified by its diagnostic code in this study, the disease severity could not be diagnosed. It was, therefore, difficult to decide whether the cataract affects the visual function or requires surgery.

## 5. Conclusions

The current findings indicate that the incidence of cataracts is lower among patients who have undergone total hysterectomy than among those who have not. Subgroup analysis further revealed that the hazard ratio for cataract formation was lower in the hysterectomy with adnexal surgery group than in the hysterectomy alone and non-hysterectomy groups. In this study, the incidence of cataracts decreased after surgical menopause, which conflicted with the effects of estrogen on cataract formation in majority of previous studies. The exact mechanism between estrogen and cataract, including surgical menopause, needs further research. This study analyzed the data of approximately 250,000 Korean Health Insurance Corporation participants and included 160,000 women in analyses adjusted for cataract risk factors. To the best of our knowledge, this is the first large-scale study to provide relevant findings that may be beneficial in understanding the effects of hysterectomy and estrogen on cataract progression.

## Figures and Tables

**Figure 1 medicina-59-01627-f001:**
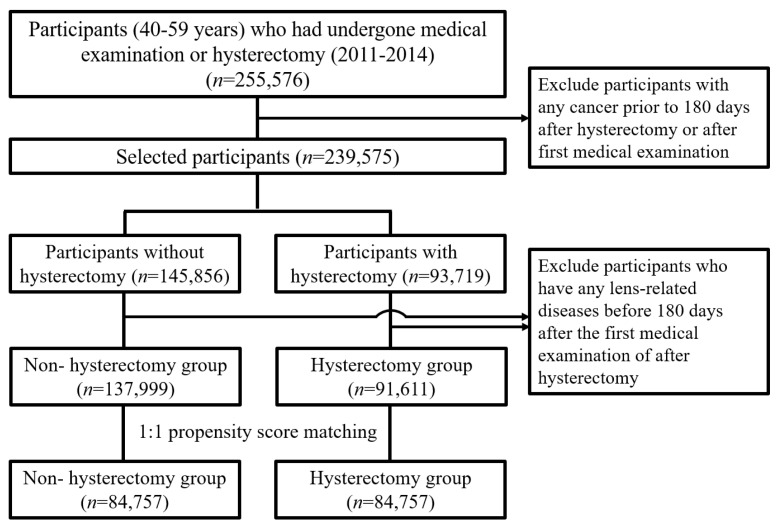
Flow chart of participant selection for the hysterectomy and control groups (National Health Insurance Database, 2007–2020).

**Figure 2 medicina-59-01627-f002:**
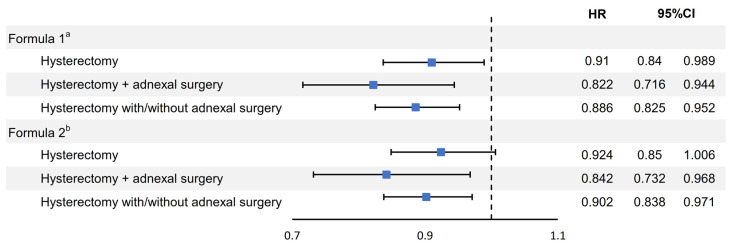
Hazard ratios for cataracts in participants with hysterectomy (National Health Insurance Database, 2007–2020). CCI, Charlson Comorbidity Index; CI, confidence interval; DM, diabetes mellitus; HR, hazard ratio; MHT, menopausal hormone therapy; SES, socioeconomic status; ^a^ HRs were adjusted for hysterectomy, age, SES, registration, CCI, parity, adnexal surgery before inclusion, hypertension before inclusion, DM before inclusion, dyslipidemia before inclusion, menopause before inclusion, and MHT before inclusion; ^b^ HRs were adjusted for hysterectomy, age, SES, registration, CCI, parity, adnexal surgery before inclusion, hypertension before inclusion, DM before inclusion, dyslipidemia before inclusion, menopause before inclusion, MHT before inclusion, and MHT after inclusion.

**Table 1 medicina-59-01627-t001:** Baseline characteristics of the participants (National Health Insurance Database, 2007–2020).

	Non-Hysterectomy	Hysterectomy	Total	*p*-Value	Standardized Difference
Number of participants	84,757	84,757	169,514		
Adnexal surgery during hysterectomy					
Hysterectomy		65,702 (77.5)	65,702 (77.5)		
Hysterectomy + adnexal surgery		19,055 (22.5)	19,055 (22.5)		
Median age (years)	46 (43–50)	47 (44–50)	46 (43–50)	<0.001	0.070
Age at inclusion (years)				<0.001	0.188
40~44	32,138 (37.9)	25,675 (30.3)	57,813 (34.1)		
45~49	29,075 (34.3)	35,957 (42.4)	65,032 (38.4)		
50~54	19,229 (22.7)	19,369 (22.9)	38,598 (22.8)		
55~59	4315 (5.1)	3756 (4.4)	8071 (4.8)		
Year at inclusion				<0.001	0.055
2011	19,963 (23.6)	21,542 (25.4)	41,505 (24.5)		
2012	20,639 (24.4)	21,256 (25.1)	41,895 (24.7)		
2013	22,852 (27)	21,592 (25.5)	44,444 (26.2)		
2014	21,303 (25.1)	20,367 (24)	41,670 (24.6)		
SES				0.953	<0.001
Mid to high SES	82,433 (97.3)	83,429 (97.3)	165,862 (97.3)		
Low SES	2324 (2.7)	2328 (2.7)	4652 (2.7)		
Region				<0.001	−0.064
Urban area	48,837 (57.6)	51,500 (60.8)	100,337 (59.2)		
Rural area	35,920 (42.4)	33,257 (39.2)	69,177 (40.8)		
CCI				0.002	0.017
0	62,805 (74.1)	62,375 (73.6)	125,180 (73.8)		
1	12,804 (15.1)	12,788 (15.1)	25,592 (15.1)		
≥2	9148 (10.8)	9594 (11.3)	18,742 (11.1)		
Parity in cohort				0.551	0.004
0	83,609 (98.6)	83,592 (98.6)	167,201 (98.6)		
1	823 (1)	818 (1)	1641 (1)		
≥2	325 (0.4)	347 (0.4)	672 (0.4)		
Adnexal surgery before inclusion				0.066	−0.009
Absent	83,577 (98.6)	83,664 (98.6)	167,241 (98.6)		
Present	1180 (1.4)	1193 (1.4)	2373 (1.4)		
Hypertension before inclusion				<0.001	0.034
Absent	71,136 (83.9)	70,048 (82.6)	141,184 (83.3)		
Present	13,621 (16.1)	14,709 (17.4)	28,330 (16.7)		
DM before inclusion				<0.001	0.023
Absent	77,460 (91.4)	76,910 (90.7)	154,370 (91.1)		
Present	7297 (8.6)	7847 (9.3)	15,144 (8.9)		
Dyslipidemia before inclusion				<0.001	0.023
Absent	67,597 (79.8)	66,817 (78.8)	134,414 (79.3)		
Present	17,160 (20.2)	17,940 (21.2)	35,100 (20.7)		
Menopause before inclusion				0.433	−0.004
Absent	75,400 (89)	75,501 (89.1)	150,901 (89)		
Present	9357 (11)	9256 (10.9)	18,613 (11)		
MHT before inclusion				<0.001	0.034
Absent	83,541 (98.6)	83,173 (98.1)	166,714 (98.3)		
Present	1216 (1.4)	1584 (1.9)	2800 (1.7)		
MHT after inclusion				<0.001	
Absent	78,696 (92.8)	71,111 (83.9)	149,807 (88.4)		
Present	6061 (7.2)	13,646 (16.1)	19,707 (11.6)		

CCI, Charlson Comorbidity Index; DM, diabetes mellitus; MHT, menopausal hormone therapy; SES, socioeconomic status. Data are expressed as the number (%) or median (25 percentile, 75 percentile).

**Table 2 medicina-59-01627-t002:** Incidence of cataracts in participants with/without hysterectomy (National Health Insurance Database, 2007–2020).

	Non-Hysterectomy	Hysterectomy		Total	*p*-Value
Number of participants	84,757	84,757		169,514	
Cataract					0.017
Absent	82,970 (97.9)	83,109 (98.1)		166,079 (98)	
Present	1787 (2.1)	1648 (1.9)		3435 (2)	
		Hysterectomy	Hysterectomy + adnexal surgery		
Cataract					0.28
Absent	82,970 (97.9)	64,467 (98.1)	18,642 (97.8)	166,079 (98)	
Present	1787 (2.1)	1235 (1.9)	413 (2.2)	3435 (2)	
MHT after inclusion					<0.001
Absent	78,696 (92.8)	56,171 (85.5)	14,940 (78.4)	149,807 (88.4)	
Present	6061 (7.2)	9531 (14.5)	4115 (21.6)	19,707 (11.6)	

Data are expressed as the number (%).

**Table 3 medicina-59-01627-t003:** Hazard ratios for cataracts in participants with hysterectomy (National Health Insurance Database, 2007–2020).

	Unadjusted		Formula 1 ^a^		Formula 2 ^b^	
	HR (95% CI)	*p*-Value	HR (95% CI)	*p*-Value	HR (95% CI)	*p*-Value
Hysterectomy						
Reference (no hysterectomy)	1 (reference)		1 (reference)		1 (reference)	
Hysterectomy alone	0.925 (0.854–1.002)	0.058	0.91 (0.837–0.989)	0.026	0.924 (0.849–1.006)	0.067
Hysterectomy + adnexal surgery	0.844 (0.737–0.967)	0.015	0.822 (0.716–0.944)	0.006	0.842 (0.732–0.968)	0.016
Hysterectomy with/without adnexal surgery	0.904 (0.843–0.968)	0.004	0.886 (0.825–0.952)	<0.001	0.902 (0.838–0.971)	0.006
MHT after inclusion					0.852 (0.735–0.988)	0.034

CCI, Charlson Comorbidity Index; CI, confidence interval; DM, diabetes mellitus; HR, hazard ratio; MHT, menopausal hormone therapy; SES, socioeconomic status; ^a^ HRs were adjusted for hysterectomy, age, SES, region, CCI, parity, adnexal surgery before inclusion, hypertension before inclusion, DM before inclusion, dyslipidemia before inclusion, menopause before inclusion, MHT before inclusion; ^b^ HRs were adjusted for hysterectomy, age, SES, region, CCI, parity, adnexal surgery before inclusion, hypertension before inclusion, DM before inclusion, dyslipidemia before inclusion, menopause before inclusion, MHT before inclusion, MHT after inclusion.

**Table 4 medicina-59-01627-t004:** Hazard ratios for cataracts in participants with/without hysterectomy according to age group (National Health Insurance Database, 2007–2020).

	40~44 Years		45~49 Years		50~54 Years		55~59 Years	
	HR (95% CI) ^a^	*p*-Value	HR (95% CI) ^a^	*p*-Value	HR (95% CI) ^a^	*p*-Value	HR (95% CI) ^a^	*p*-Value
Cataract (Formula 1) ^a^								
Reference (no hysterectomy)	1 (reference)		1 (reference)		1 (reference)		1 (reference)	
Hysterectomy	1.106 (0.88–1.389)	0.389	0.998 (0.662–1.156)	0.982	0.833 (0.727–0.955)	0.009	0.872 (0.684–1.112)	0.271
Hysterectomy + adnexal surgery	0.793 (0.498–1.263)	0.329	0.919 (0.698–1.212)	0.551	0.877 (0.703–1.093)	0.242	0.724 (0.532–0.987)	0.041
Hysterectomy with/without adnexal surgery	1.038 (0.845–1.275)	0.724	0.981 (0.861–1.117)	0.77	0.845 (0.751–0.95)	0.005	0.813 (0.669–0.988)	0.038
Cataract (Formula 2) ^b^								
Reference (no hysterectomy)	1 (reference)		1 (reference)		1 (reference)		1 (reference)	
Hysterectomy	1.038 (0.845–1.275)	0.724	0.983 (0.863–1.12)	0.795	0.845 (0.751–0.95)	0.005	0.813 (0.669–0.988)	0.038
Hysterectomy + adnexal surgery	1.106 (0.88–1.389)	0.387	1.003 (0.866–1.161)	0.971	0.833 (0.727–0.955)	0.009	0.872 (0.684–1.112)	0.271
Hysterectomy with/without adnexal surgery	0.793 (0.498–1.263)	0.329	0.914 (0.693–1.205)	0.523	0.877 (0.703–1.093)	0.242	0.724 (0.532–0.987)	0.041

CCI, Charlson Comorbidity Index; CI, confidence interval; DM, diabetes mellitus; HR, hazard ratio; MHT, menopausal hormone therapy; SES, socioeconomic status. ^a^ HRs were adjusted for hysterectomy, age, SES, region, CCI, parity, adnexal surgery before inclusion, hypertension before inclusion, DM before inclusion, dyslipidemia before inclusion, menopause before inclusion, MHT before inclusion. ^b^ HRs were adjusted for hysterectomy, age, SES, region, CCI, parity, adnexal surgery before inclusion, hypertension before inclusion, DM before inclusion, dyslipidemia before inclusion, menopause before inclusion, MHT before inclusion, MHT after inclusion.

## Data Availability

HIRA permits analysis only on closed servers, and only the anonymized resultant tables, figures, and numbers can be taken outside the server. Therefore, researchers are fundamentally unable to specify each patient, and there are no possible disadvantages for the participants. In accordance with the Bioethics and Safety Act of South Korea, these participants were not obliged to provide informed consent. Therefore, raw data could not be provided.

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
