# Peer review of "Hysterectomy for Benign Indications and Risk of Cataract Formation in South Korean Women"

_medicina, 2023, doi:10.3390/medicina59091627_

Round 1

Reviewer 1 Report

In this article, the authors investigated the effects of hysterectomy on the risk of
lens opacity in Korean women.
This study is well-designed, enjoyable, and contains novelties, although it is exact
the practical utility is unclear.
 Minor Remarks:

 - In the Introduction part, there is little information about the role and mechanism of action of estrogen in the development of cataracts.

- The authors summarized our result in the discussion part: "In this large-scale, long-term study of Korean Health Insurance Corporation data for Korean women, we observed that hysterectomy decreased the risk of cataract formation compared to that observed in the control group, but the difference was subtle.” "subtle" is not a statistical or scientific category. Please, modify it!

  - I feel a discrepancy between in Introduction "estrogen may protect against the development of cataracts" and later "Estrogen not having significant effects on the development of cataracts". Resolve or discuss the contradiction!

- The MHT could be an important part of the manuscript. Generally, after a hysterectomy used just estrogen supplementation without progesterone. It would be interesting to examine the frequency of cataracts as a drug therapy as well.

- Formulate a "message" about the practical, therapeutic importance of this study in the conclusion part of the manuscript!

Author Response

In this article, the authors investigated the effects of hysterectomy on the risk of ens opacity in Korean women.
This study is well-designed, enjoyable, and contains novelties, although it is exact the practical utility is unclear. 

We appreciate the reviewer’s precise comments regarding our manuscript.

 Minor Remarks:

 - In the Introduction part, there is little information about the role and mechanism of action of estrogen in the development of cataracts.

Thank you for your comment. Introduction has been amended after reviewing the Reviewer’s comments. We deeply appreciate the reviewer’s comments, which have helped improve our manuscript. These detailed comments have been truly helpful.

- The authors summarized our result in the discussion part: "In this large-scale, long-term study of Korean Health Insurance Corporation data for Korean women, we observed that hysterectomy decreased the risk of cataract formation compared to that observed in the control group, but the difference was subtle.” "subtle" is not a statistical or scientific category. Please, modify it!

Thank you for your comment. The sentence has been modified in accordance with the Reviewer’s comment.

- I feel a discrepancy between in Introduction "estrogen may protect against the development of cataracts" and later "Estrogen not having significant effects on the development of cataracts". Resolve or discuss the contradiction!

Thank you for your comment. The sentence has been modified in accordance with the Reviewer’s comment. We greatly appreciate your kind suggestions.

- The MHT could be an important part of the manuscript. Generally, after a hysterectomy used just estrogen supplementation without progesterone. It would be interesting to examine the frequency of cataracts as a drug therapy as well.

Thank you for your accurate point. The authors discussed about this suggestion and agreed unanimously. And we added in the discussion section of the manuscript

- Formulate a "message" about the practical, therapeutic importance of this study in the conclusion part of the manuscript!

The authors totally agreed on this suggestion. The authors added therapeutic importance of this study in the conclusion section.

Reviewer 2 Report

The author constructed a well-designed retrospective cohort study to investigate the effects hysterectomy and estrogen on the incidence of cataracts in Korean women.

The research design is rigorous, and the manuscript is well described. After adjusting for numerous variables (age, BMI, SES, CCI, parity, age at menarche, age at menopause, smoking, alcohol consumption, physical exercise, and period since menopause), and 1:1 propensity score matching, the study revealed that the incidence of cataracts is lower among patients who have undergone total hysterectomy than among those who have not.

The conclusion is interesting, but the author is suggested to enrich the speculation about the mechanism. Additionally, the author should also highlight the clinical and public implications of these results, otherwise readers may be confused or even make wrong decisions.

Author Response

The author constructed a well-designed retrospective cohort study to investigate the effects hysterectomy and estrogen on the incidence of cataracts in Korean women.

The research design is rigorous, and the manuscript is well described. After adjusting for numerous variables (age, BMI, SES, CCI, parity, age at menarche, age at menopause, smoking, alcohol consumption, physical exercise, and period since menopause), and 1:1 propensity score matching, the study revealed that the incidence of cataracts is lower among patients who have undergone total hysterectomy than among those who have not.

 The authors appreciate the reviewer’s kind comments about our manuscript.

The conclusion is interesting, but the author is suggested to enrich the speculation about the mechanism. Additionally, the author should also highlight the clinical and public implications of these results, otherwise readers may be confused or even make wrong decisions.

Thank you for your accurate point. The authors totally agreed on this suggestion. The conclusion section  has been amended after reviewing the Reviewer’s comments. We deeply appreciate the reviewer’s comments,
